# No general stability conditions for marine ice-sheet grounding lines in the presence of feedbacks

Olga V. Sergienko [1 ✉]

The "marine ice-sheet instability" hypothesis continues to be used to interpret the observed mass loss from the Antarctic and Greenland ice sheets. This hypothesis has been developed for conditions that do not account for feedbacks between ice sheets and environmental conditions. However, snow accumulation and the ice-sheet surface melting depend on the surface temperature, which is a strong function of elevation. Consequently, there is a feedback between precipitation, atmospheric surface temperature and ice-sheet surface elevation. Here, we investigate stability conditions of a marine-based ice sheet in the presence of such a feedback. Our results show that no general stability condition similar to one associated with the "marine ice-sheet instability" hypothesis can be determined. Stability of individual configurations can be established only on a case-by-case basis. These results apply to a wide range of feedbacks between marine ice sheets and atmosphere, ocean and lithosphere.

[1] Program in Atmospheric and Oceanic Sciences, Princeton University, Princeton, NJ, USA. ✉email: osergien@princeton.edu

The observed mass loss from the Antarctic and Greenland ice sheets associated with the retreat of their grounding lines[1–3] continues to be explained[2–4] by the "marine ice-sheet instability" hypothesis[5], even though many recent studies have questioned its validity[6–9]. This hypothesis proposed by Weertman[5], suggests that a marine ice sheet that resides on a bed below sea level and slopes towards its interior is "inherently unstable". He further suggested that "A stable ice sheet can occur if the bed slopes away from the center of the ice sheet." Weertman arrived to this conclusion by considering an unconfined marine ice sheet flowing into an unconfined ice shelf that gains its mass by accumulating snow at its surface with the accumulation rate constant in space and time. In later studies[10,11], a similar ice-sheet configuration was considered and the same stability condition was derived by means of a linear stability analysis[12].

Ice sheets interact with all other components of the Earth system—atmosphere, ocean, sea ice, lithosphere, land surface, and biosphere. These interactions give rise to a number of feedbacks[13]; one of them is a feedback between the net accumulation/ablation rate and the ice-sheet height[14–16]. The net accumulation/ablation rate at a given location on the surface of the ice-sheet is determined by the difference between snow accumulation and surface melting (ablation). If the annual snow precipitation exceeds annual surface melt (if melting occurs at all) this location experiences net accumulation; if the reverse is true, it experiences net ablation. Both processes, snow precipitation, and surface melting, depend on in situ atmospheric conditions. Snow precipitation depends on the atmospheric moisture content and the surface ablation rate is determined by the surface energy balance. The latter involves a number of processes, e.g., absorption of short- and long-wave radiation, reflection due to albedo, wind, and many others. Both the snow precipitation rate and ablation rate strongly depend on the surface elevation[17]. Such a dependence produces a feedback between the surface elevation and the net accumulation/ablation rate[15].

While several studies[14,15,18] have considered the effects of a feedback between the surface elevation and behavior of land-based ice sheets, all existing studies of stability of marine ice sheets[5,10–12,19–21] did not take into account feedbacks between the external conditions and the ice-sheet characteristics. It is, thus, not a priori clear whether the results of these studies remain valid in the presence of feedbacks.

Here we show that the previously derived stability conditions of unconfined marine ice sheets that disregard the feedback between net accumulation/ablation rate and the surface elevation do not generally hold if the feedback is taken into account. Furthermore, there is no general stability condition which is determined by properties of steady-state configurations at the grounding line. Stability of specific configurations can be determined only individually, on a case-by-case basis. These results apply to other feedbacks in interactions of marine ice sheets with other components of the climate system—ocean (the dependence of sub-ice-shelf melting on ice thickness), continental crust and lithosphere (changes in bed elevation due to erosion/deposition, glaciostatic adjustment and or/changes in the local sea level).

## Results

**Accumulation/ablation rate dependence on surface temperature.** Precipitation is determined by the atmospheric moisture content. A good proxy for the moisture content is the saturation vapor pressure, governed by the Clausius-Clapeyron equation, which in turn, depends on the surface temperature[17]. Ablation depends on the (near) surface temperature in two ways—the latter determines whether surface melting occurs (if the surface temperature is above the freezing point), and it determines the magnitude of the surface melting, the ablation rate. Although there are many complex processes that govern interactions between the ice-sheet surface and atmosphere, the atmospheric surface temperature can be used as a sole proxy parameter to estimate both precipitation and surface melting, and consequently the net accumulation/ablation rate.

Atmospheric temperature is a strong function of elevation: it decreases according to the lapse rate as elevation increases. As a result, the net accumulation/ablation rate implicitly depends on the ice-sheet surface elevation via the surface temperature.

Recent developments in the regional atmospheric modeling for polar regions have allowed for more accurate representation of the near-surface atmospheric processes[22,23]. In the absence of the direct continent-wide long-term observations, this has made results of these models' simulations a useful tool for analysis of the polar atmospheric conditions. For example, results of simulations of the regional atmospheric model, MAR, have been used to construct a parameterization of changes in the net accumulation/ablation due to changes in the surface elevation of Greenland Ice Sheet based on a probabilistic approach[24].

In this study, the results of MAR simulations, for both Antarctica[25] and Greenland[26] are used to construct an empirical relationship between the net accumulation/ablation rate and the surface temperature. The simulated accumulation/ablation rates, $\dot{a}$, as a function of the annual mean surface temperatures for the RCP 8.5 scenario for Antarctica (1980–2100) and Greenland (2006–2100) are shown in Fig. 1a. Temperatures simulated in this scenario have a wide range of magnitudes and include significantly higher values than those observed or simulated for the present day conditions (Supplementary Figs. 1 and 2). The empirically fitted expression for $\dot{a}$ as a function of the annual

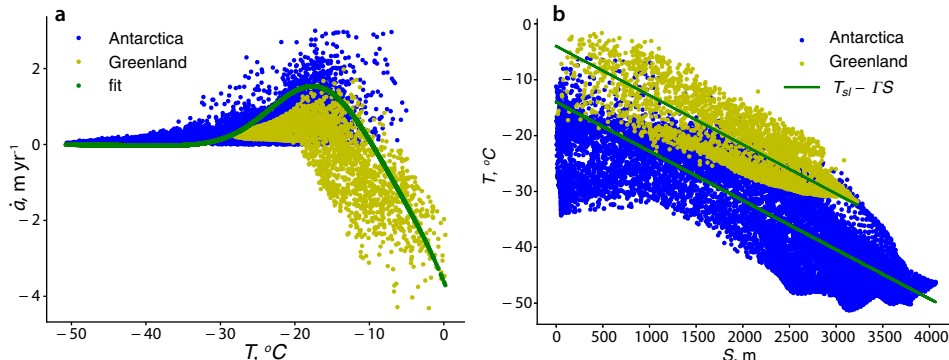

**Fig. 1 Results MAR simulations Antarctica (blue dots) 1980–2100 (ref. [25]), Greenland (yellow dots) 2006–2100 (ref. [26]). a** Net accumulation/ablation rate $\dot{a}$(m/yr) as function of the surface temperature $T$ (°C); green dots indicate fit described by expression (1). **b** Surface temperature $T$ (°C) versus ice sheet surface elevation $S$ (m); green lines are expression (2) with $\Gamma = 9.8$ °C/km, $T_{sl} = -15$ °C for Antarctica, and $T_{sl} = -5$ °C for Greenland, respectively.

mean surface temperature $T$ is described by Eq. (1) (see the "Methods" section).

Results of MAR simulations (Fig. 1b) show a linear decrease of the annual mean surface temperature with elevation supporting the relationship (2). Also, these results indicate that simulated temperatures around Greenland (yellow dots) are consistently warmer at all elevations than around Antarctica (blue dots), and substantially warmer at sea level $T_s$ ($S = 0$). Consequently, considering the accumulation/ablation rate of Antarctica and Greenland together, and simulated for the high-end emission scenario, allows construction of an empirical relationship for a wider range of surface temperatures and surface elevations that can simultaneously capture the net accumulation and ablation regimes.

Relating the net accumulation/ablation to the surface temperature has other advantages, in addition to the physical basis of the chosen relationship. As (Fig. 1b) indicates, depending on the climate conditions temperature at the same surface elevations can be different; relating the net accumulation/ablation rate to the surface temperature and not surface elevation removes this ambiguity. A simple dependence on the annual mean surface temperature at sea level $T_{sl}$ makes the empirical relationship (1) together with (2) an ideal tool for theoretical and conceptual studies that aim to establish fundamental aspects of interactions between ice sheets and atmosphere, and also for paleo-climate studies as it eliminates the need to compute precipitation and the surface melting separately. "Methods" section and Supplementary Fig. 3 illustrate how sea-level temperature $T_{sl}$ controls the behavior of $\dot{a}(T_S(S))$, and demonstrate that the empirical relationships (1) and (2) can capture regimes characteristic of MAR simulations for Antarctica (blue dots) and Greenland (yellow dots in Fig. 1a).

**The effect of a feedback on stability of marine ice sheets.** A steady-state configuration of an unconfined marine ice sheet can be described by a solution of the problem (3) described in the "Methods" section, which is the same as a model used in ref. [10]. An example of a steady-state marine ice sheet, for which accumulation/ablation rate varies with the surface elevation according to relationships (1) and (2), is shown in Fig. 2a. Its grounding line $x_g$ is located on the up-sloping (or "retrograde") bed. According to both a stability condition derived by Schoof[12] and a more complex one that accounts for the bed curvature and the gradient of the accumulation/ablation rate $\dot{a}_x$ derived by Sergienko and Wingham[20], this grounding line position is stable.

These stability analyses, however, have been performed under the assumption that $\dot{a}$ varies only along the length of the ice sheet, as illustrated in Fig. 2b, and does not depend on any characteristics of the ice sheet, e.g., its surface elevation. To determine whether this steady-state configuration is stable in the presence of a feedback between the accumulation/ablation rate and the surface elevation, Eqs. (1) and (2), a numerical time-dependent model simulation, in which the grounding line position is perturbed from its steady-state position, is performed. The numerical model solves the time-variant problem described by Eqs. (3a), (3c)–(3e). The mass-balance evolves in time according to Eq. (4), which accounts for the feedback between the net accumulation/ablation rate and the surface elevation i.e., $\dot{a} = \dot{a}(T_S(S))$. (Details of the time-dependent simulations are described in Supplementary Methods 1). Snapshots of the time-evolving ice-sheet surface and the corresponding spatial distributions of $\dot{a}(T_S(S))$ are shown in Supplementary Fig. 6. Evolution of the simulated grounding line position is shown in Fig. 3a. The grounding line advances beyond its steady-state position (marked by the black dashed line) indicating that the steady-state position is unstable. This contradicts stability conditions[12,20] that do not account for feedbacks.

However, if $\dot{a}$ is treated as a function of the position, and not of the surface elevation, i.e., $\dot{a} = \dot{a}(x)$ as shown in Fig. 2b, the simulated advance of the grounding line from a similarly perturbed position terminates at its steady-state location, indicating that in this case the steady-state configuration is stable (Fig. 3b), which is in agreement with traditional analytical stability conditions[12,20] that do not account for feedbacks. This experiment illustrates that it is the presence of the feedback that changes stability of the grounding line from stable if $\dot{a} = \dot{a}(x)$ to unstable if $\dot{a} = \dot{a}(T_S(S))$.

To shed light on such a different behavior of the grounding lines of ice sheets, which accumulation/ablation rates vary either with the surface elevation or only with the distance along the ice sheet, we analyze a simplified problem (5) used to establish a more complex stability condition[20] for the case of the net accumulation/ablation rate being a function of the position, $\dot{a} = \dot{a}(x)$. A linear stability analysis for the case of the feedback between the net accumulation/ablation rate and the surface elevation $\dot{a} = \dot{a}(T_S(S))$ follows the same standard approach[20], and considers small, time-dependent perturbations around the steady-state solutions, e.g., $x_g = \hat{x}_g + \tilde{x}_g(t)$, where $\hat{x}_g$ is a steady-state position of the grounding line and $\tilde{x}_g(t)$ is a small time-variant perturbation. The evolution of the grounding line in the corresponding linearized perturbation problem (Supplementary Methods 2) is described by $\tilde{x}_g(t) = \tilde{x}_g(t=0)e^{\Lambda t}$, where $\Lambda$ is an eigenvalue. A stability condition for the case of $\dot{a} = \dot{a}(x)$ relies on properties of the eigenfunctions of the perturbation problem[27] and is determined by the sign of the largest eigenvalue—either the fastest growing, in the case of an unstable grounding line position, or the slowest decaying in the case of a stable grounding

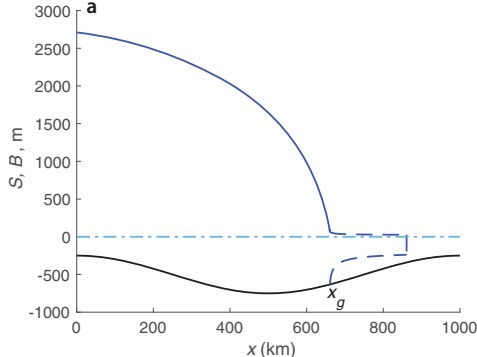

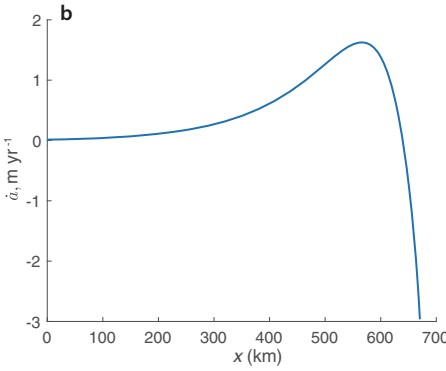

**Fig. 2 Steady-state ice-sheet configuration and accumulation/ablation rate. a** Surface elevation $S$ (blue line), bed elevation $B$ (black line). **b** Accumulation/ablation rate $\dot{a}$ (m yr$^{-1}$) computed using Eq. (1) with $T_{sl} = -4$ °C.

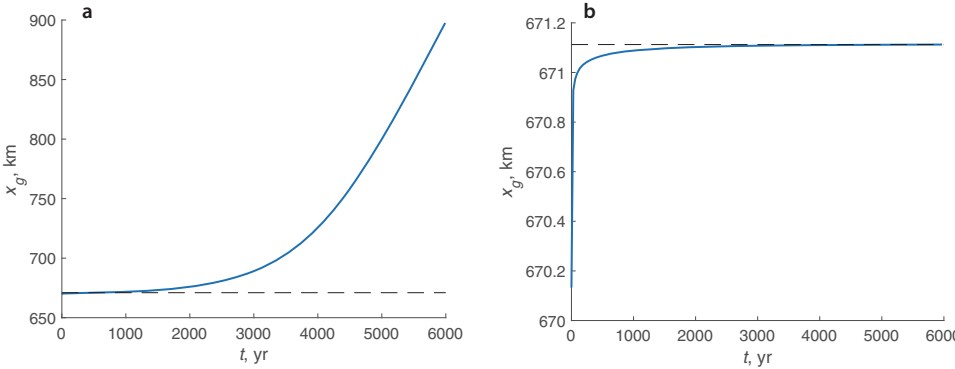

**Fig. 3 Evolution of the perturbed grounding line positions. a** For the case of $\dot{a}(T)$ described by Eq. (1). **b** For the case of $\dot{a}$ treated as a function of $x$ (Fig. 2b). Dashed lines show steady-state position of the grounding lines. Note different vertical scales in panels **a** and **b**.

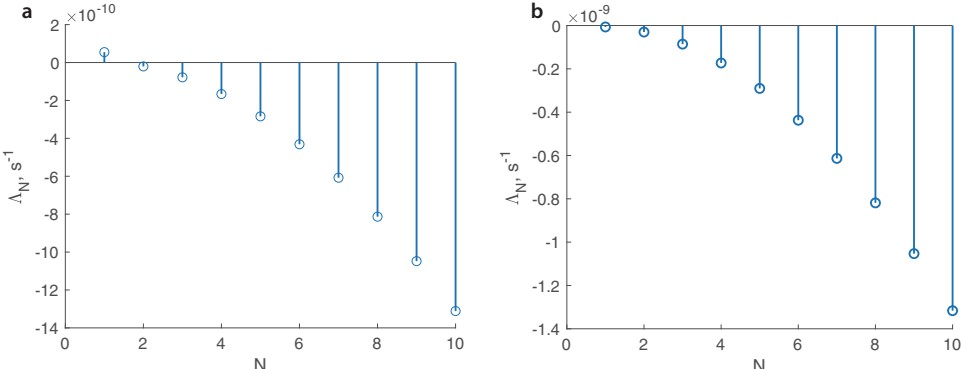

**Fig. 4 Eigenvalues of the perturbed eigenvalue problem. a** For the case of $\dot{a}(T)$ described by Eq. (1). **b** For the case of $\dot{a}(x)$ treated as a function of $x$ shown in Fig. 2b.

line position[20]. In the case of $\dot{a} = \dot{a}(T_S(S))$ no inferences about either the sign or magnitude of eigenvalues can be made; and the perturbation eigenvalue problem has to be solved explicitly to determine the largest eigenvalue. This implies that there is no general stability condition that depends on properties of the steady-state ice-sheet configuration in the presence of the feedback between the net accumulation/ablation rate and the surface elevation. As Supplementary Eq. (14) indicates, the absence of the general stability condition in the presence of the feedbacks is insensitive to specifics of the functional form of the empirical relationship described by Eqs. (1) and (2).

Results of a numerical solution of the linear-perturbation eigenvalue problem are illustrated in Fig. 4a that shows the first ten eigenvalues. The first eigenvalue is positive indicating that the small perturbations from the steady-state configuration grow with time, i.e., the steady-state configuration and its grounding line position is unstable. This is in agreement with the results of the temporal evolution of the grounding line perturbed from its steady-state location (Fig. 3a). Eigenvalues for the case of $\dot{a}(x)$ are shown in Fig. 4b. The largest eigenvalue is negative, indicating that the small perturbations from the steady-state solution decay in time, and the steady-state solution is stable. This is again in agreement with the results of the temporal evolution of the grounding line perturbed from its steady-state location (Fig. 3b).

## Discussion
In order to investigate how feedbacks may affect stability conditions of marine ice sheets, we have constructed an empirical relationship between the surface temperature, which varies linearly with elevation according to the lapse rate, and the net

accumulation/ablation rate. The constructed formulation uses the fundamental relationships between the atmospheric surface temperature and precipitation (Clausius-Clapeyron equation) and the surface temperature and ice-sheet surface melting (energy balance). Expressed in terms of temperature at the sea level, $T_{sl}$ this formulation can be an efficient, physically based parameterization of the net accumulation/ablation for paleo-climate simulations.

Numerical and analytical analyses of the marine ice-sheet stability performed with this empirical relationship produce results markedly different from those of previous analyses that did not account for any feedbacks. Stability conditions derived for the cases with no feedbacks between the ice-sheet characteristics (e.g., the ice-sheet surface elevation) and other parameters (e.g., the net accumulation/ablation rate) do not hold in general if such feedbacks are present. Moreover, in the demonstration of this considered here, there are no general stability conditions that depend on characteristics of the steady-state configurations at the grounding line. In the presence of feedbacks, stability of a particular ice-sheet steady-state configuration can be determined only by either numerically simulating evolution of this configuration perturbed from its steady state, or by solving a perturbation eigenvalue problem to determine the largest eigenvalue, which sign indicate stability (all eigenvalues are negative) or instability (at least one eigenvalue is positive).

A physical interpretation of these results is the following: If the net accumulation/ablation rate depends on the surface elevation then the specific details of such a dependence (e.g., how $\dot{a}$ varies with $S$), together with the characteristics of the steady-state configuration, determine whether this steady-state configuration is stable or unstable. Changes in net accumulation/ablation rate

caused by changes in surface elevation due to small perturbations from steady-state configurations could suppress or amplify these perturbations, and consequently stabilize or destabilize the grounding line.

The presented analysis can be applied to other processes that include feedbacks. If sub-ice-shelf melting occurs at the grounding line, the melting rate is determined by the pressure melting point, which depends on the ice thickness. This introduces a feedback between ice thickness and sub-ice-shelf melting at the grounding line. If the bed elevation varies due to, for instance, glaciostatic adjustment and erosion/deposition processes, or due to changes in the local sea level, the surface elevation and temperature vary as a result of these processes. In all these circumstances the presented results remain valid.

A number of processes that have feedbacks with the ice-sheet characteristics are in play on the Antarctic and Greenland ice sheets[13]. Consequently, their stability conditions are not determined by the bed slopes at their grounding lines alone as suggested by the traditional "marine ice-sheet instability hypothesis"[5]. Instead, stability conditions depend on the interplay between buttressing (if the ice shelves are confined), basal conditions (bed elevation, slope, curvature and sliding properties) and feedbacks between the ice sheet configuration and atmosphere, ocean, and lithosphere interactions.

## Methods

**Empirical relationship between $\dot{a}$ and $S$.** The fitting (green) line in Fig. 1a has the following expression

$$\dot{a}(T_S) = a_1 \exp\left[-\frac{(T_S - T_0)^2}{2\sigma^2}\right] - a_2 \exp\left[-2\frac{T_S - T_0}{T_0}\right] \quad (1)$$

where $a_1$, $a_2$, $T_0$, and $\sigma$ are fitting parameters and $T_S$ is the temperature at the surface elevation $S$, which is

$$T_S = T_{sl} - \Gamma S \quad (2)$$

where $T_{sl}$ is the temperature at sea level and $\Gamma$ is the lapse rate. In this study, we use the dry adiabatic lapse rate $\Gamma = 9.8\,°C/km$. In general, surface temperature at the sea level depends on insolation, which is function of latitude. In this study, we disregard this dependence and treat sea-level surface temperature $T_{sl}$ as constant.

The magnitude of the sea-level temperature $T_{sl}$ controls whether an ice sheet can experience net ablation or it experiences only net accumulation. The steady-state configuration shown in Fig. 2a has been computed with $T_{sl} = -4\,°C$; the corresponding net ablation/accumulation rate shown in Fig. 2b as a function of distance, in Supplementary Fig. 4a as a function of surface elevation and in Supplementary Fig. 4b as a function of surface temperature experiences net ablation (negative values of $\dot{a}$) at the low elevations close to the grounding line. These results indicate that this relationship captures MAR simulations for Greenland in warmer climate of the RCP 8.5 scenario (yellow dots in Fig. 1a). For colder sea-level temperature $T_{sl} = -10\,°C$, a steady-state configuration computed with Eqs. (1) and (2) does not experience net ablation (Supplementary Fig. 3). As Supplementary Fig. 3d illustrates, for this value of $T_{sl}$ the constructed empirical relationship captures the behavior of $\dot{a}$ shown in Fig. 1a obtained in MAR simulations for Antarctica (blue dots).

**Marine ice-sheet model.** We use the model of ref. [10]. As in many previous studies of unconfined marine ice sheets flowing into unconfined ice shelves, the momentum balance of the ice shelf can be integrated analytically and the problem is reduced to one of the marine ice-sheet grounded parts only[28]. In a steady-state it is

$$2\left(A^{-1/n}h\left|u_x\right|^{1/n-1}u_x\right)_x - \tau_b - \rho g h S_x = 0 \quad \text{for } x_d \le x \le x_g, \quad (3a)$$

$$(uh)_x = \dot{a}(T_S), \quad (3b)$$

$$S_x = 0, \quad u = 0 \quad \text{at } x = x_d, \quad (3c)$$

$$2A^{-1/n}h\left|u_x\right|^{1/n-1}u_x = \frac{1}{2}\rho g' h^2 \quad \text{at } x = x_g, \quad (3d)$$

$$h = -\frac{\rho_w}{\rho}B \quad \text{at } x = x_g. \quad (3e)$$

where $u(x)$ is the depth-averaged ice velocity, $h(x)$ ice thickness, $B(x)$ is bed elevation (negative below sea level and positive above sea level), $S = h + B$ is the

surface elevation, $A^{-1/n}$ is the ice stiffness parameter (assumed to be constant), $n$ is the exponent of Glen's flow law, $g$ is the acceleration due to gravity, $\tau_b = C|u|^{m-1}u$ is basal shear with constant parameters $C$ and $m = 1/n$, $x_d$ and $x_g$ are positions of the ice divide (taken here to be $x_d = 0$) and the grounding line, respectively, and $\dot{a}(T_S)$ is described by (1), with $T_S$ described by (2); $g'$ is the reduced gravity defined as $g' = \delta g$, where $\delta = \frac{\rho_w - \rho}{\rho_w}$ is the buoyancy parameter.

In the case of temporal evolution of the ice-sheet configuration, the mass balance (3b) takes the form

$$h_t + (uh)_x = \dot{a}(T_S(S)) \quad (4)$$

**Simplified model.** An approximation of (3) with (4) instead of (3b) written in terms of the ice thickness $h$ and the ice flux $q = uh$ is

$$-C\frac{|q|^{m-1}q}{h^m} - \rho g h S_x = 0 \quad \text{for } x_d \le x \le x_g, \quad (5a)$$

$$h_t + (uh)_x = \dot{a}, \quad (5b)$$

$$S_x = 0, \quad u = 0 \quad \text{at } x = x_d, \quad (5c)$$

$$\frac{C}{\rho g}q^{m+1} + qh^{m+1}B_x = \left(\frac{A^{\frac{1}{n}}}{4}\rho g'\right)^n h^{n+m+3} - (\dot{a} - h_t)h^{m+2} \quad \text{at } x = x_g, \quad (5d)$$

$$h = -\frac{\rho_w}{\rho}B \quad \text{at } x = x_g. \quad (5e)$$

To determine stability conditions, a linear stability analysis of (5) is performed by considering small perturbations from the steady-state solutions. The details of the linear stability analysis in the presence of a feedback between the net accumulation/ablation rate and the ice-sheet surface, i.e., $\dot{a} = \dot{a}(T_S(S))$ are described in Supplementary Methods 2; the details of the linear stability analysis in the absence of such a feedback i.e., $\dot{a} = \dot{a}(x)$ are described in ref. [20].

## Data availability

Outputs of MAR simulations for Antarctica[25] are available at the Zenodo database[29] https://zenodo.org/record/4459259 and for Greenland[26] at ftp://ftp.climato.be/fettweis/MARv3.5/Greenland/. Data shown in the figures have been deposited in the Zenodo database[30] under accession code https://zenodo.org/record/5724988.

## Code availability

Numerical models used in this study have been deposited in the Zenodo database[30] under accession code https://zenodo.org/record/5724988.

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

## Acknowledgements
This study is supported by award NA18OAR4320123 from the National Oceanic and Atmospheric Administration, U.S. Department of Commerce. The statements, findings, conclusions, and recommendations are those of the authors and do not necessarily reflect the views of the National Oceanic and Atmospheric Administration, or the U.S. Department of Commerce.

## Author contributions
O.S. designed and performed the study and wrote the manuscript.

## Competing interests
The author declares no competing interests.
