## [Peer Review File · Nature Communications]

No general stability conditions for marine ice-sheet grounding lines in the presence of feedbacksReviewers' Comments:

Reviewer #1:

Remarks to the Author:

Please see attached report

Review of “Stability of marine ice sheets in the presence of the surface-elevation feedbacks”, by
O. Sergienko

September 2021

1 Overview

This is a nice analysis of the role surface mass balance feedbacks could play in marine ice sheet stability. The author presents a straightforward and compelling demonstration of how feedbacks from changes in surface mass balance due to changes in surface elevation can play an important role in the stability of marine ice sheets. I think this adds to our understanding of the mechanics of marine ice sheet stability and is a useful addition to the literature. I would support publication after some issues are addressed.

One issue with this work is that it seems to jump directly from the idealized analyses of Schoof circa 2007 and ignores progress made in the last decade or so in our understanding of the stability of marine ice sheets in more-realistic configurations, particularly the importance of ice-shelf buttressing as a stabilizing process (for example [2]). In other words, it's already understood that the idealized unconfined marine ice sheet examples of Weertman and Schoof are a cartoon which provides insight, but are not the final word on marine ice sheet stability.

Also, surface elevation/SMB feedback and its impact on stability is already well known in the case of the Greenland Ice Sheet independent of specific marine ice sheet considerations (see for example [1] and [3]). While this work breaks new ground by extending it to the case of marine ice sheets, this work should at least mention the previous work demonstrating the importance of surface elevation/SMB feedback in other ice sheet contexts.

While these two lines of research (and others) don't directly impact the results of this work, they do place it into the larger body of work which is seeking to understand ice sheet stability in less idealized configurations; the author should mention both of these to help place this work in context for the reader.

Based on Figure 1, it's not clear at all that you can create a single fit encompassing Antarctica and Greenland. The two sets of data points really don't appear to have much in common. You'll notice that the vast majority of the

Greenland points (which I appreciate are conveniently colored green!) lie under the fit line, for one thing. Also, from what I can see, the Antarctic data points show no sign of the fall-off with increasing temperature that the Greenland data displays. Instead what I see is a fairly consistent floor at 0 for the blue-colored data points accompanied by a widening range of outputs as temperature increases. On the other hand, the Greenland data looks like a very nice constant-thickness band around the mean fit. I think you need to either be able to justify this combined fit, or separate out the Antarctic and Greenland cases and treat them separately. It seems likely that the stability story due SMB feedbacks may be different for the two ice sheets, which would be an interesting result as well. The plots in the Supplemental Material seem to reinforce this – the two datasets in Figure S3 seem to be quite independent of each other; the Greenland data seems well-correlated by the best-fit line, while the Antarctic data looks like it might be better represented by a parabolic fit given what appears to be a concave-down curvature.

My biggest issue with this work as presented, however, is that I don't understand the finite-perturbation stability example described in line 88 and afterwards. How do you perturb the grounding line by 1 km upstream? It's not clear in the main body, and the description in line 39 in the supplemental material doesn't give me any more clarity: "The time-variant simulations are performed using steady-state solutions as initial conditions, in which the grounding line positions are 1 km upstream of their steady-state positions." This seems to imply that you're starting with a steady-state solution which has a grounding line 1 km upstream from its steady-state position, which seems contradictory to me.

My assumption is that perturbing the grounding line 1 km upstream entails a thinner ice sheet than the reference, which would imply lower surface elevation \rightarrow lower surface temperature \rightarrow *reduced* (or more negative) \dot{a} . It's not obvious to me how a perturbation which reduced the SMB contribution would lead to the substantial advance beyond the steady-state position shown in figure 3(a); rather, I'd expect sustained thinning and retreat. However, one could imagine that a perturbation which *advanced* the grounding line (thickening the ice sheet and thereby increasing \dot{a}) could lead to such a runaway advance. It's obvious to me that I'm missing something here, which implies that a clearer description of the experiment is warranted.

Since you're using a fixed vertical temperature profile (I think) for this experiment, I think it would also help clarify things if you included a plot of \dot{a} as a function of elevation (essentially substituting $T_s(z)$ into the expression for $\dot{a}(T_s)$ to produce $\dot{a}(z)$ as used in this experiment); this could be presented as a part of the supplemental material, but I think it would help readers interpret your experiment.

2 Specific points

1. line 44: "atmosphere moisture content" \rightarrow "atmospheric"?

2. line 47: I think that “(the surface temperature...” should be “(if the surface temperature...”
3. line 58: “models” → “models’ ”
4. line 59: “of of”
5. line 63: “for RCP 8.5 scenario...” → “for the RCP 8.5 scenario...”
6. line 73: “pale-climate” → “paleo-climate”
7. line 81: As mentioned earlier, I think it’s worth mentioning either here or somewhere in the introduction that this simple unconfined ice sheet example neglects ice shelf buttressing, which appears to be important to many (if not most) Antarctic marine ice sheets.
8. line 83: “which accumulation/ablation...” → “for which...”
9. line 85: I don’t think there should be commas after “both” and “Schoof”
10. line 95: missing space after “elevation”
11. Figure 3: Probably obvious to you, but it would be helpful to a casual reader to call out the fact that the vertical scales are different between (a) and (b).
12. Figure 3 caption: “Dashed lines show steady-state positions...” – I’d suggest changing “positions” to “position” to reinforce that it’s the same steady-state grounding-line position in both plots. Otherwise “positions” implies that they’re different, which is reinforced at first glance by the fact that they’re in different vertical locations on the plots; this confused me for a bit until I looked at the vertical axes.
13. line 170: As I mentioned earlier, stability conditions depend on these things *and* buttressing effects in ice shelves in realistic configurations
14. Reference 3: “Antarctic Ice Sheet” isn’t capitalized.

References

- [1] T. L. Edwards, X. Fettweis, O. Gagliardini, F. Gillet-Chaulet, H. Goelzer, J. M. Gregory, M. Hoffman, P. Huybrechts, A. J. Payne, M. Perego, S. Price, A. Quiquet, and C. Ritz. Probabilistic parameterisation of the surface mass balance–elevation feedback in regional climate model simulations of the Greenland ice sheet. *The Cryosphere*, 8(1):181–194, 2014.
- [2] G. H. Gudmundsson. Ice-shelf buttressing and the stability of marine ice sheets. *The Cryosphere*, 7(2):647–655, 2013.
- [3] A. Levermann and R. Winkelmann. A simple equation for the melt elevation feedback of ice sheets. *The Cryosphere*, 10(4):1799–1807, 2016.

Reviewer #2:

Remarks to the Author:

This is a work of unusually high quality, a solid theoretical work addressing an important question, and likely to be cited for many years to come.

My main criticism of the manuscript is that the author could state more clearly the importance of the work. While reading it was initially quite confused. Based on the title and the abstract I was expecting some extension of Weertman's 1961 work on the stability of ice sheets in the presence of mass-balance-altitude feedbacks. After reading the paper I realized that this the manuscript addresses the importance of upstream (ie upstream from the grounding line) mass-balance for the grounding-line stability. By including the mass-balance feedback, as done in this work, the mass-balance and the geometry become linked. Personally, I did not find this to be very surprising, but good scientific work does not have to be surprising. It should address an important question and provide solid answers, and this the manuscript does.

Below I've added some of the comments I wrote while reading the manuscript. Possibly the author might find it useful to see how I was initially struggling to understand the focus of the manuscript. Maybe this will help in restructuring the paper.

I found the last sentence of the abstract very confusion: "These results indicate that the observed mass-loss from the present-day ice sheets, which surface processes include a number of feedbacks, cannot be attributed to marine ice-sheet instability in its traditional form." What does this sentence mean? That the currently observed mass loss of the Antarctic Ice Sheet (AIS) cannot be attributed to the marine ice sheet instability (MISI)? But is anyone claiming that it is? The author will fully understand (better than most) that grounding-line retreat is not as such evidence of an instability. The first sentence in the main text that observed mass loss from AIS and GIS is 'traditionally' explained by the MISI is news to me.

I'm also puzzled by the way the research question is framed. Yes mass-balance depends on surface-elevation feedback and this feedback can give rise to an unstable behavior (e.g. Weertman) but is the key thing here not just the relationship between the change in upstream integrated mass balance compared to the change in ice-flux as the grounding line advances? As such, this is not inherently a process that depends on mass-balance-altitude feedback. In Weertman 1961, in contrast, unstable states arise because of the mass-balance-altitude feedback. I guess the thinking might be that for the grounding-line advance to be unstable, the flux across the grounding line becomes progressively less as the grounding line advances while the integrated surface mass balance upstream generally increases (hence unstable). Not sure how the mass-balance-elevation feedback can be expected to fundamentally change this picture. When the grounding line advances the elevation also increases and then with that the surface mass balance, so if anything, the imbalance is further increased.

Initially, I quite frankly found the discussion and the link to MISI a distraction. But after having read the manuscript, I think this is due to the way the initial part (title and abstract) is written. The mass-balance-elevation feedback is important in its own as it can give rise to a genuine instability that has nothing to do with the MISI. There are numerous minor further distractions in the manuscript. Now need really, for example, to discuss degree-day factors or computational difficulties with resolving seasonal cycles.

The example given in Fig. 2 is presumably stable because the mass balance becomes strongly negative as the grounding line advances, compensation for the decrease in ice flux as the ice-thickness at the grounding line becomes smaller. This is not particularly surprising, but it is worthwhile to point out that the stability is not independent of the surface-mass balance, and I guess this is the key message of the paper. But still, it is not the mass-balance-altitude-feedback as such that changes the stability but the relative change in surface mass balance (a) as the grounding line position (x) changes, and the change in grounding line flux (q), i.e what matters here is how da/dx and dq/dh

dh/dx changes, where h is the thickness at the grounding line. In the example give in figure 2 (unbuttressed ice shelf) $dh/dx < 0$ and $dq/dh > 0$ so $dq/dh dh/dx < 0$, so if $da/dx > 0$ we have an instability. In Fig2. da/dx appears to become sufficiently strongly negative past about $x=550\text{km}$ for a steady state position to be found at about $x=650\text{km}$.

This is most likely just me having difficulties understanding some of the steps of the instability analysis. I got the linearization but had difficulties understanding how the steady-state was generated. Apparently, this was done numerically. I was then expecting some stability analysis similar to Mulder et al (2018) but did not see a description of how the (steady-state) solution branch was traced. How can the author be sure that the steady-state solution was not simple stable because of the way it was found? Were continuum methods used to trace the branch and find potential bifurcation points? I suspect this is just me not grasping the link between the numerical solution and the perturbation approach, but I would have expected the stability to have been assessed by looking at the eigenvalues of the (numerical) Jacobian as typically done in numerical work. This would then have required the mass-balance altitude feedback to be included in the Jacobian, and it was not clear to me if that had been done.

Q: What are the noteworthy results?

A: Stability of grounding-lines is not independent of upstream accumulation. So, for example even for unconfined ice-stream/ice-shelf systems, the grounding line stability can not be judged on local bed slope alone as often assumed. If furthermore the surface mass balance depends on surface elevation, it becomes difficult to make any general statements about GL stability.

Q: Will the work be of significance to the field and related fields? How does it compare to the established literature? If the work is not original, please provide relevant references.

A: This is of importance to our understanding of the qualitative behavior of large ice-sheets, and of the West-Antarctic Ice Sheet (WAIS) in particular.

Q: Does the work support the conclusions and claims, or is additional evidence needed?
Yes, the work supports the conclusions, and does so in quite frankly in a much more rigorous way than typical for all the impending-doom stories I read in Nature manuscripts about dynamics of large ice sheets (MICI comes to the mind).

Q: Are there any flaws in the data analysis, interpretation, and conclusions? Do these prohibit publication or require revision?

A: I do feel the work could be presented better and sold better. I was initially quite confused about the focus of the paper and thought it was related to the instability that can happen due to mass-balance altitude feedback, whereas the manuscript focuses on how upstream mass balance affects grounding-line stability, which is a rather different mechanism.

Q: Is the methodology sound? Does the work meet the expected standards in your field?

A: It is of similar quality as found in the better papers of the field such as The Cryosphere and The Journal of Glaciology.

Q: Is there enough detail provided in the methods for the work to be reproduced?

A: Yes, although I personally did not fully follow the link between the numerical steady-state solution and the analytical perturbation analysis.

Overall, I really enjoyed reading this work. Refreshing to read a submission that is based on solid mathematical and physical understanding of the subject. Allowing myself to make a slightly flippant remark towards the end, I am however not sure if this manuscript is the kind of work that Nature publishes. Maybe the author could just add a sentence about the (bonkers) marine ice cliff instability, that should do the job.

Papers referred to:

Weertman, J. (1961). Stability of ice-age ice sheets. *Journal of Geophysical Research*, 66(11), 3783–3792. <https://doi.org/10.1029/JZ066i011p03783>

Mulder, T. E., Baars, S., Wubs, F. W., & Dijkstra, H. A. (2018). Stochastic marine ice sheet variability. *Journal of Fluid Mechanics*, 843, 748–777. <https://doi.org/10.1017/jfm.2018.148>

Reviewer #3:

Remarks to the Author:

Sergienko et al., have investigated the stability of an idealised marine ice sheet with and without a surface mass balance-elevation feedback. The manuscript presents simulations of an idealised ice sheet profile (as is common in theoretical explorations of marine ice sheet instability), combined with an idealised (but crucially, elevation dependent) surface mass balance forcing. Simulations show that with this feedback, there is no general stability state. This is an important finding, and progresses fundamental glaciology. While it is widely acknowledged that the Marine Ice Sheet Instability hypothesis interacts with other forcings of ice sheet change, this manuscript is the first to provide a robust mathematical exploration of these interactions.

The paper is clear and well-written, and all the figures are also clear. The mathematical description is complete, sensibly organised, and well explained. In my view, this work represents a significant advance in fundamental glaciology, and I recommend that this manuscript is accepted with minor revisions. I have appended some specific points for revision below.

Major points

- Ln 70-79. I think it is very sensible to use an idealised climate forcing as described here. But I think that justification should be clearer/reworked in the manuscript. I am not sure that a PDD scheme is costly enough to warrant this simpler approach. Primarily I am curious how sensitive your simulation results are to the surface mass balance parameterisation.

Minor points

- Ln 47. “- the latter...”. I don’t follow this sentence, could it be cleared up?
- Ln 63. Why only use the RCP8.5 scenario results? I’d imagine (but don’t know!) that Figure 1 would look very similar with the data from all the scenarios included.
- Ln 73. “pale-climate” > palaeo/paleo
- Ln 74. I would say the usually a PDD scheme is used in palaeo studies because of its relatively small computational cost (i.e. versus a more desirable but more costly energy balance approach). As you say, the main disadvantage of a PDD scheme is the unknown Degree-Day factors.
- Ln 182 and 201. “ref” typo

Reviewer 1

Overview

R1: *This is a nice analysis of the role surface mass balance feedbacks could play in marine ice sheet stability. The author presents a straightforward and compelling demonstration of how feedbacks from changes in surface mass balance due to changes in surface elevation can play an important role in the stability of marine ice sheets. I think this adds to our understanding of the mechanics of marine ice sheet stability and is a useful addition to the literature. I would support publication after some issues are addressed.*

I am grateful to the Reviewer for their time and effort reading my manuscript, providing thoughtful and useful suggestions and for their positive evaluation of the manuscript.

R1: *One issue with this work is that it seems to jump directly from the idealized analyses of Schoof circa 2007 and ignores progress made in the last decade or so in our understanding of the stability of marine ice sheets in more-realistic configurations, particularly the importance of ice-shelf buttressing as a stabilizing process (for example [1]). In other words, it's already understood that the idealized unconfined marine ice sheet examples of Weertman and Schoof are a cartoon which provides insight, but are not the final word on marine ice sheet stability.*

The Reviewer is absolutely correct about the recent improvements in understanding the stability of marine ice sheets. However, many publications continue to use Weertman's and Schoof's explanations to interpret the observed retreat of the grounding line. The cited references [2–4] explicitly do that. To reflect the recent progress in understanding of the stability of marine ice sheets, the abstract (line 1) and introduction (lines 13-15) have been modified and now include references to recent studies on the marine ice-sheet stability [1, 5–7].

R1: *Also, surface elevation/SMB feedback and its impact on stability is already well known in the case of the Greenland Ice Sheet independent of specific marine ice sheet considerations (see for example [8] and [9]). While this work breaks new ground by extending it to the case of marine ice sheets, this work should at least mention the previous work demonstrating the importance of surface elevation/SMB feedback in other ice sheet contexts. While these two lines of research (and others) don't directly impact the results of this work, they do place it into the larger body of work which is seeking to understand ice sheet stability in less idealized configurations; the author should mention both of these to help place this work in context for the reader.*

A clarifying statement about the existing studies of the feedback between surface elevation and

accumulation/ablation rate and references mentioned by the Reviewer and other ones have been added to introduction (line 36).

R1: *Based on Figure 1, it's not clear at all that you can create a single fit encompassing Antarctica and Greenland. The two sets of data points really don't appear to have much in common. You'll notice that the vast majority of the Greenland points (which I appreciate are conveniently colored green!) lie under the fit line, for one thing. Also, from what I can see, the Antarctic data points show no sign of the fall-off with increasing temperature that the Greenland data displays. Instead what I see is a fairly consistent floor at 0 for the blue-colored data points accompanied by a widening range of outputs as temperature increases. On the other hand, the Greenland data looks like a very nice constant-thickness band around the mean fit. I think you need to either be able to justify this combined fit, or separate out the Antarctic and Greenland cases and treat them separately. It seems likely that the stability story due SMB feedbacks may be different for the two ice sheets, which would be an interesting result as well. The plots in the Supplemental Material seem to reinforce this – the two data sets in Figure S3 seem to be quite independent of each other; the Greenland data seems well-correlated by the best-fit line, while the Antarctic data looks like it might be better represented by a parabolic fit given what appears to be a concave-down curvature.*

To address this comment, figure 1b has been moved from the Supplementary Information (originally figure 3) to the main text. The Reviewer is correct that the simulated temperatures for Antarctica (blue dots) and Greenland (yellow) appear to be separated in this plot. This is due to the fact that the simulated Greenland temperatures are consistently warmer than Antarctic ones by approximately 10°C at all elevations. The simulated Antarctic temperatures are not sufficiently warm to result in surface melting (figure 1a). On the other hand, the simulated Greenland temperatures are not as cold as the Antarctic ones, and the Greenland Ice Sheet is not as high as the Antarctic Ice Sheet. Consequently simulated surface conditions of both ice sheets are needed to construct an empirical relationship between the net accumulation/ablation and the surface temperature that can encompass a wide range of cold temperatures and warm temperatures that capture different regimes of low accumulation corresponding to cold temperatures; net accumulation corresponding to warmer temperatures; and net ablation corresponding to even warmer temperatures. This is also the reason to consider the RCP 8.5 scenario. Simulations for other RCP scenarios result in colder temperatures and lower or non-existent net ablation. A clarifying statement has been added to lines 78-84, 86-96.

A new figure has been added to Supplementary Information (Fig. 5). It shows the steady-state configuration (panel a) and the corresponding accumulation/ablation rate as a function of distance

(panel b), surface elevation (panel c) and surface temperature (panel d) computed for sea-level temperature $T_{sl}=-10^{\circ}\text{C}$. As Fig. 5d illustrates, for this value of T_{sl} the constructed empirical relationship captures a regime obtained with MAR simulations for Antarctica (blue dots in Fig. 1a). Results of simulations with $T_{sl}=-4^{\circ}\text{C}$ shown in Fig. 2 and Supplementary Fig.3b indicate that this relationship captures MAR simulations for Greenland in warmer climate of the RCP 8.5 scenario. Clarifying statements describing these results have been added to Methods section, lines 205-215.

R1: *My biggest issue with this work as presented, however, is that I don't understand the finite-perturbation stability example described in line 88 and afterwards. How do you perturb the grounding line by 1 km upstream? It's not clear in the main body, and the description in line 39 in the supplemental material doesn't give me any more clarity: "The time-variant simulations are performed using steady-state solutions as initial conditions, in which the grounding line positions are 1 km upstream of their steady-state positions." This seems to imply that you're starting with a steady-state solution which has a grounding line 1 km upstream from its steady-state position, which seems contradictory to me. My assumption is that perturbing the grounding line 1 km upstream entails a thinner ice sheet than the reference, which would imply lower surface elevation \rightarrow lower surface temperature \rightarrow reduced (or more negative) \dot{a} . It's not obvious to me how a perturbation which reduced the SMB contribution would lead to the substantial advance beyond the steady-state position shown in figure 3(a); rather, I'd expect sustained thinning and retreat. However, one could imagine that a perturbation which advanced the grounding line (thickening the ice sheet and thereby increasing \dot{a}) could lead to such a runaway advance. It's obvious to me that I'm missing something here, which implies that a clearer description of the experiment is warranted.*

An ambiguous description of the perturbation time-dependent simulations has been clarified in the main text (lines 109-115) and in the Supplementary Methods 2 (lines 43-53). The perturbation simulations use u and h obtained in steady-state simulations as initial conditions. At $t = 0$ the grounding line position is set 1 km upstream of its steady-state position, this ice-sheet configuration is no longer in a steady state and the grounding line begins to advance. In contrast to the case of $\dot{a} = \dot{a}(x)$, in which the grounding line advances to its steady-state position and remains there (Fig. 3b), in the case of $\dot{a} = \dot{a}(T_s(S))$, the ice sheet thickens and \dot{a} increases leading to the runaway advance. A new figure in Supplementary Information (fig. 4) illustrates evolution of the unstable ice-sheet configuration and corresponding changes in \dot{a} . The Reviewer is absolutely correct about the thickening ice sheet, increasing \dot{a} and runaway advance.

R1: *Since you're using a fixed vertical temperature profile (I think) for this experiment, I think it would also help clarify things if you included a plot of \dot{a} as a function of elevation (essentially*

substituting $T_s(z)$ into the expression for $\dot{a}(T_s)$ to produce $\dot{a}(z)$ as used in this experiment); this could be presented as a part of the supplemental material, but I think it would help readers interpret your experiment.

The profile of \dot{a} as a function of S is shown in fig. 3a (fig. 4 in the original version) of Supplementary Information. This profile is the same for steady-state and time-evolving experiments.

Specific points

R1: 1. line 44: “atmosphere moisture content” → “atmospheric”?

Corrected.

R1: 2. line 47: I think that “(the surface temperature...” should be “(if the surface temperature...”

Corrected.

R1: 3. line 58: “models” → “models’ ”

Corrected.

R1: 4. line 59: “of of”

Corrected.

R1: 5. line 63: “for RCP 8.5 scenario...” → “for the RCP 8.5 scenario...”

Corrected.

R1: 6. line 73: “pale-climate” → “paleo-climate”

Corrected.

R1: 7. line 81: *As mentioned earlier, I think it’s worth mentioning either here or somewhere in the introduction that this simple unconfined ice sheet example neglects ice shelf buttressing, which appears to be important to many (if not most) Antarctic marine ice sheets.*

This and additional references have been added to introduction.

R1: 8. line 83: “which accumulation/ablation...” → “for which...”

Corrected.

R1: 9. line 85: I don’t think there should be commas after “both” and “Schoof”

Corrected.

R1: 10. line 95: missing space after “elevation”

Corrected.

R1: 11. Figure 3: Probably obvious to you, but it would be helpful to a casual reader to call out the fact that the vertical scales are different between (a) and (b).

A note about different vertical scales on panels **a** and **b** has been added.

R1: Figure 3 caption: “Dashed lines show steady-state positions...” – I’d suggest changing “positions” to “position” to reinforce that it’s the same steady-state grounding-line position in both plots. Otherwise “positions” implies that they’re different, which is reinforced at first glance by the fact that they’re in different vertical locations on the plots; this confused me for a bit until I looked at the vertical axes.

Corrected.

R1: 13. line 170: As I mentioned earlier, stability conditions depend on these things and buttressing effects in ice shelves in realistic configurations

This sentence has been modified.

R1: 14. Reference 3: “Antarctic Ice Sheet” isn’t capitalized.

Corrected.

Reviewer 2

R2: *This is a work of unusually high quality, a solid theoretical work addressing an important question, and likely to be cited for many years to come.*

I am grateful to the Reviewer for their appreciation of this study.

R2: *My main criticism of the manuscript is that the author could state more clearly the importance of the work. While reading it was initially quite confused. Based on the title and the abstract I was expecting some extension of Weertman's 1961 work on the stability of ice sheets in the presence of mass-balance-altitude feedbacks. After reading the paper I realized that this the manuscript addresses the importance of upstream (i.e. upstream from the grounding line) mass-balance for the grounding-line stability. By including the mass-balance feedback, as done in this work, the mass-balance and the geometry become linked. Personally, I did not find this to be very surprising, but good scientific work does not have to be surprising. It should address an important question and provide solid answers, and this the manuscript does.*

To address the Reviewer's criticism, the title, abstract and introduction have been modified and state more clearly what problem has been considered and how it is different from the earlier analysis by Weertman 1961 [10] and others.

R2: *I found the last sentence of the abstract very confusion: "These results indicate that the observed mass-loss from the present-day ice sheets, which surface processes include a number of feedbacks, cannot be attributed to marine ice-sheet instability in its traditional form." What does this sentence mean? That the currently observed mass loss of the Antarctic Ice Sheet (AIS) cannot be attributed to the marine ice sheet instability (MISI)? But is anyone claiming that it is? The author will fully understand (better than most) that grounding-line retreat is not as such evidence of an instability. The first sentence in the main text that observed mass loss from AIS and GIS is 'traditionally' explained by the MISI is news to me.*

The first mentioned sentence has been removed from the abstract. The first sentences in abstract (line 1) and introduction (lines 13-15) have been modified to reflect that MISI continues to be used to explain mass-loss from the Antarctic and Greenland ice sheets. The cited references are recent papers in high-profile journals that explicitly do that. Introduction has been modified to reflect a growing body of studies that question MISI and its validity.

R2: *I'm also puzzled by the way the research question is framed. Yes mass-balance depends*

on surface-elevation feedback and this feedback can give rise to an unstable behavior (e.g. Weertman) but is the key thing here not just the relationship between the change in upstream integrated mass balance compared to the change in ice-flux as the grounding line advances? As such, this is not inherently a process that depends on mass-balance-altitude feedback. In Weertman 1961, in contrast, unstable states arise because of the mass-balance-altitude feedback. I guess the thinking might be that for the grounding-line advance to be unstable, the flux across the grounding line becomes progressively less as the grounding line advances while the integrated surface mass balance upstream generally increases (hence unstable). Not sure how the mass-balance-elevation feedback can be expected to fundamentally change this picture. When the grounding line advances the elevation also increases and then with that the surface mass balance, so if anything, the imbalance is further increased.

Clarifying statements about the main finding – the absence of the general stability condition in the presence of a feedback between net accumulation/ablation – have been added to introduction (lines 41-49).

R2: *Initially, I quite frankly found the discussion and the link to MISI a distraction. But after having read the manuscript, I think this is due to the way the initial part (title and abstract) is written. The mass-balance-elevation feedback is important in its own as it can give rise to a genuine instability that has nothing to do with the MISI. There are numerous minor further distractions in the manuscript. Now need really, for example, to discuss degree-day factors or computational difficulties with resolving seasonal cycles.*

The title and abstract have been modified. References to the Positive Degree Day method and degree-day factors have been removed.

R2: *The example given in Fig. 2 is presumably stable because the mass balance becomes strongly negative as the grounding line advances, compensation for the decrease in ice flux as the ice-thickness at the grounding line becomes smaller. This is not particularly surprising, but it is worthwhile to point out that the stability is not independent of the surface-mass balance, and I guess this is the key message of the paper. But still, it is not the mass-balance-altitude-feedback as such that changes the stability but the relative change in surface mass balance (a) as the grounding line position (x) changes, and the change in grounding line flux (q), i.e what matters here is how da/dx and dq/dh dh/dx changes, where h is the thickness at the grounding line. In the example give in figure 2 (unbuttressed ice shelf) $dh/dx < 0$ and $dq/dh > 0$ so $dq/dhdh/dx < 0$, so if $da/dx > 0$ we have an instability. In Fig2. da/dx appears to become sufficiently strongly negative*

past about $x=550\text{km}$ for a steady state position to be found at about $x=650\text{km}$.

The Reviewer is correct. The example in Fig.2 is stable because of the net ablation in the vicinity of the grounding line. The linear stability analysis described in Supplementary Methods 1 explicitly demonstrates that in the presence of the feedback between net accumulation/ablation, how \dot{a} changes with S determines whether a steady state is stable or unstable. It is due to this dependence it is not possible to derive a general stability condition.

R2: *This is most likely just me having difficulties understanding some of the steps of the instability analysis. I got the linearization but had difficulties understanding how the steady-state was generated. Apparently, this was done numerically. I was then expecting some stability analysis similar to Mulder et al (2018) but did not see a description of how the (steady-state) solution branch was traced. How can the author be sure that the steady-state solution was not simple stable because of the way it was found? Were continuum methods used to trace the branch and find potential bifurcation points? I suspect this is just me not grasping the link between the numerical solution and the perturbation approach, but I would have expected the stability to have been assessed by looking at the eigenvalues of the (numerical) Jacobian as typically done in numerical work. This would then have required the mass-balance altitude feedback to be included in the Jacobian, and it was not clear to me if that had been done.*

The steady-state solutions were determined by solving numerically the steady-state problem (Methods eqns (3)), i.e. with $h_t=0$. The fact that the obtained solutions were indeed steady states has been verified by using these solutions as initial conditions for time-dependent simulations with all parameters identical to the steady-state simulations. In these time-dependent simulations the ice-sheet configurations did not change and remained the same as the initial, steady-state configurations.

To establish the stability of the obtained steady-state configurations two perturbation methods have been used. The first one is the time-dependent numerical simulations in which the steady-state configurations were perturbed. The steady-state solutions for u and h have been used as initial conditions for these simulations. At $t=0$ the grounding line positions were set 1 km upstream of their steady-state locations; because of this displacement of the grounding lines, the ice-sheet configurations were no longer in a steady state and started to evolve. In both cases, the grounding lines started to advance to their steady-state positions. In the case of $\dot{a} = \dot{a}(x)$ the grounding line reached its steady state location and remained there (Fig. 3b). In the case of $\dot{a}(T_S(S))$ the grounding line advanced beyond its steady-state location due to thickening of the

ice sheet and increase in the accumulation rate. A new figure 4 in Supplementary Information illustrates this behaviour. Clarifying statements have been added to the main text (lines 109-115) and to the Supplementary Methods 2 (lines 43-53).

The second method is a linear stability analysis of a simplified problem (5) described in Methods section. A perturbation problem (Supplementary Methods 1) has been constructed by assuming linear response of the system to small perturbations. This linear perturbation problem includes both the momentum (eqn. (3a)) and mass (eqn. (3b)) balances. It can be written in a Sturm-Liouville form (eqn. (7a)). The sign of eigenvalues of this Sturm-Liouville problem (eqn. 7) determine stability of a steady state for which a linear perturbation problem is considered. Analysis of this Sturm-Liouville problem indicates that it is not possible to determine the sign of the eigenvalues if net ablation/accumulation depends on the surface elevation (*e.g.* there is perturbation in \dot{a} due to perturbations in S or rather in h , as bed elevation b is assumed to be constant in time) This is described by eqn. (13) and lines 27-32 of Supplementary Information. In addition to this analysis, the perturbation problem (eqn. 7) has been solved numerically to determine eigenvalues shown in fig. 4 using steady-state solutions of h and u as \hat{h} and \hat{u} . This method, based on Sturm-Liouville problem, allows to establish stability of a specific steady state. In contrast, a method used by [11] relies on construction of a Lyapunov problem and a bifurcation diagram of a specific parameter, (the ice-stiffness parameter A in the case considered by [11]).

Both methods used in this study to establish stability conditions of steady states are independent of how these steady states were obtained.

R2: *Overall, I really enjoyed reading this work. Refreshing to read a submission that is based on solid mathematical and physical understanding of the subject. Allowing myself to make a slightly flippant remark towards the end, I am however not sure if this manuscript is the kind of work that Nature publishes. Maybe the author could just add a sentence about the (bonkers) marine ice cliff instability, that should do the job.*

I appreciate the irony of this comment.

Reviewer 3

R3: *Sergienko et al., have investigated the stability of an idealised marine ice sheet with and without a surface mass balance-elevation feedback. The manuscript presents simulations of an idealised ice sheet profile (as is common in theoretical explorations of marine ice sheet instability), combined with an idealised (but crucially, elevation dependent) surface mass balance forcing. Simulations show that with this feedback, there is no general stability state. This is an important finding, and progresses fundamental glaciology. While it is widely acknowledged that the Marine Ice Sheet Instability hypothesis interacts with other forcings of ice sheet change, this manuscript is the first to provide a robust mathematical exploration of these interactions.*

I thank the Reviewer for their time and effort reviewing my manuscript.

R3: *The paper is clear and well-written, and all the figures are also clear. The mathematical description is complete, sensibly organised, and well explained. In my view, this work represents a significant advance in fundamental glaciology, and I recommend that this manuscript is accepted with minor revisions. I have appended some specific points for revision below.*

Major points

• *Ln 70-79. I think it is very sensible to use an idealised climate forcing as described here. But I think that justification should be clearer/reworked in the manuscript. I am not sure that a PDD scheme is costly enough to warrant this simpler approach. Primarily I am curious how sensitive your simulation results are to the surface mass balance parameterisation.*

A justification to use an idealized parameterization has been added to lines 85-96. The reference to PDD scheme has been removed. The main result of the analysis – the absence of a general stability condition in the presence of the feedback between net accumulation/ablation rate and surface elevation – is independent of a specific form and parameters of the established empirical relationship. A clarifying comment has been added to Supplementary Methods 1 (lines 32-35). A new Supplementary figure 5 illustrates a steady-state configuration (panel a) and corresponding variability of \dot{a} with the distance (panel b), elevation (panel c) and surface temperature (panel d) for sea-level temperature $T_{sl} = -10^\circ\text{C}$. This Supplementary figure 5 and figure 2 of the main text together with Supplementary figure 3 give a sense how the simulation results depend on the net accumulation/ablation rate parameterization for warmer and colder sea-level temperatures.

Minor points

R3: • *Ln 47. “- the latter...”. I don’t follow this sentence, could it be cleared up?*

This sentence has been clarified.

R3: • Ln 63. *Why only use the RCP8.5 scenario results? I'd imagine (but don't know!) that Figure 1 would look very similar with the data from all the scenarios included.*

For lower emission scenarios the net ablation is much lower even at low elevations in Greenland due to colder surface temperatures. The goal of this empirical relationship is to cover as wide range of surface temperatures as possible to be suitable for a wide range climate scenarios. The results of the RCP 8.5 scenario for high elevations/low temperatures/low accumulation rates Antarctic simulations are very similar to other scenarios. For this reason (high surface temperature/high ablation rates at low elevations in Greenland and very low temperatures and accumulation rates a high elevations in Antarctica) the RCP 8.5 has been used.

R3: • Ln 73. *“pale-climate” & palaeo/paleo*

Corrected.

R3: • Ln 74. *I would say the usually a PDD scheme is used in palaeo studies because of it's relatively small computational cost (i.e. versus a more desirable but more costly energy balance approach). As you say, the main disadvantage of a PDD scheme is the unknown Degree-Day factors.*

This part has been removed.

R3: • Ln 182 and 201. *“ref” typo*

Here “ref” is to identify corresponding references.

References

- [1] G. H. Gudmundsson. Ice-shelf buttressing and the stability of marine ice sheets. *The Cryosphere*, 7(2):647–655, 2013. <https://doi.org/10.5194/tc-7-647-2013> doi:10.5194/tc-7-647-2013.
- [2] E. J. Rignot. Fast Recession of a West Antarctic Glacier. *Science*, 281(5376):549–551, 1998. <https://doi.org/10.1126/science.281.5376.549> doi:10.1126/science.281.5376.549.
- [3] Andrew Shepherd, Helen Amanda Fricker, and Sinead Louise Farrell. Trends and connections across the Antarctic cryosphere. *Nature*, 558(7709):223–232, 2018. <https://doi.org/10.1038/s41586-018-0171-6> doi:10.1038/s41586-018-0171-6.

- [4] Shfaqat A. Khan, Anders A. Bjørk, Jonathan L. Bamber, Mathieu Morlighem, Michael Bevis, Kurt H. Kjær, Jérémie Mouginot, Anja Løkkegaard, David M. Holland, Andy Aschwanden, Bao Zhang, Veit Helm, Niels J. Korsgaard, William Colgan, Nicolaj K. Larsen, Lin Liu, Karina Hansen, Valentina Barletta, Trine S. Dahl-Jensen, Anne Sofie Søndergaard, Beata M. Csatho, Ingo Sasgen, Jason Box, and Toni Schenk. Centennial response of Greenland’s three largest outlet glaciers. *Nature Communications*, 11(1):5718, 2020. <https://doi.org/10.1038/s41467-020-19580-5> doi:10.1038/s41467-020-19580-5.
- [5] C. Schoof, A. D. Devis, and T. V. Popa. Boundary layer models for calving marine outlet glaciers. *The Cryosphere*, 11(5):2283–2303, 2017. <https://doi.org/10.5194/tc-11-2283-2017> doi:10.5194/tc-11-2283-2017.
- [6] M. Haseloff and O. V. Sergienko. The effect of buttressing on grounding line dynamics. *Journal of Glaciology*, 64(245):417–431, 2018. <https://doi.org/10.1017/jog.2018.30> doi:10.1017/jog.2018.30.
- [7] Samuel S. Pegler. Suppression of marine ice sheet instability. *Journal of Fluid Mechanics*, 857:648–680, 2018. <https://doi.org/10.1017/jfm.2018.742> doi:10.1017/jfm.2018.742.
- [8] T. L. Edwards, X. Fettweis, O. Gagliardini, F. Gillet-Chaulet, H. Goelzer, J. M. Gregory, M. Hoffman, P. Huybrechts, A. J. Payne, M. Perego, S. Price, A. Quiquet, and C. Ritz. Probabilistic parameterisation of the surface mass balance–elevation feedback in regional climate model simulations of the Greenland ice sheet. *The Cryosphere*, 8(1):181–194, 2014. <https://doi.org/10.5194/tc-8-181-2014> doi:10.5194/tc-8-181-2014.
- [9] A. Levermann and R. Winkelmann. A simple equation for the melt elevation feedback of ice sheets. *The Cryosphere*, 10(4):1799–1807, 2016. <https://doi.org/10.5194/tc-10-1799-2016> doi:10.5194/tc-10-1799-2016.
- [10] J. Weertman. Stability of ice-age ice sheets. *Journal of Geophysical Research (1896-1977)*, 66(11):3783–3792, 1961. <https://doi.org/10.1029/JZ066i011p03783> doi:https://doi.org/10.1029/JZ066i011p03783.
- [11] T. E. Mulder, S. Baars, F. W. Wubs, and H. A. Dijkstra. Stochastic marine ice sheet variability. *Journal of Fluid Mechanics*, 843:748–777, 2018. <https://doi.org/10.1017/jfm.2018.148> doi:10.1017/jfm.2018.148.

Reviewers' Comments:

Reviewer #1:

Remarks to the Author:

See attached review

Review of Revision 1 of “No general stability conditions for marine ice-sheet grounding lines in the presence of feedbacks” by O. Sergienko

December 2021

1 Overview

The author has done a nice job of responding to most of the points I raised in my review, mostly by clarifying points which I felt were unclear and adding context about recent improvements in our understanding of marine ice sheet stability beyond the simple retrograde-bed criterion, and I thank her for her responsiveness. The work remains a nice demonstration of how marine ice sheet stability (and indeed all ice sheet stability) must be considered in a holistic way which includes feedbacks with the climate system. I continue to support publication once my request below is addressed.

Apologies if I’m missing something, but I find that I’m no clearer on how the numerical experiment described starting on line 109 is constructed; in particular, the details of how the grounding line position is perturbed continue to elude me, both in the main text and in the supplementary material.

In essence, the author determines the stable steady-state thickness profile (and resulting surface elevation) for the specified scenario; let’s call them $h_{ss}(x)$ and $S_{ss}(x)$. $h_{ss}(x)$ is at the flotation thickness at the steady-state GL position x_{SS} . The model is then initialized at time $t = 0$ with a thickness ($h(x, t = 0)$) and surface profile ($S(x, t = 0)$) which result in an initial grounding-line position that’s 1 km upstream of x_{SS} . I would like the author to provide the details of how she constructed the initial perturbed state $h(x, t = 0)$ from the steady-state profile $h_{SS}(x)$ – I presume that $h(x, t = 0) < h_{SS}(x)$ in order to move the grounding line upstream. Since the dynamic response demonstrated in this example likely depends at least somewhat on the initial perturbed state, this detail will be necessary for others to repeat this experiment.

2 Specific Points

A few minor suggestions:

1. line 17: “its it interior” → “its interior”

2. line 31: “atmosphere” → “atmospheric”.
3. line 33: no comma after “Both”
4. line 41: “That previously” → “that the previously...”
5. line 82: “allows to construct...” → “allows construction of” or “allows us to construct”
6. line 104: “this grounding line position is stable” – Did you mean “unstable” (since it’s on a retrograde slope in Figure 1)?

Reviewer #2:
None

Reviewer 1

1 Overview

R1: *The author has done a nice job of responding to most of the points I raised in my review, mostly by clarifying points which I felt were unclear and adding context about recent improvements in our understanding of marine ice sheet stability beyond the simple retrograde-bed criterion, and I thank her for her responsiveness. The work remains a nice demonstration of how marine ice sheet stability (and indeed all ice sheet stability) must be considered in a holistic way which includes feedbacks with the climate system. I continue to support publication once my request below is addressed.*

I am grateful to the Reviewer for their time and effort reading the revised version of the manuscript and for their continuing support of its publication.

R1: *Apologies if I'm missing something, but I find that I'm no clearer on how the numerical experiment described starting on line 109 is constructed; in particular, the details of how the grounding line position is perturbed continue to elude me, both in the main text and in the supplementary material. In essence, the author determines the stable steady-state thickness profile (and resulting surface elevation) for the specified scenario;*

The steady-state profiles do not have to be stable. They are obtained by solving an optimization problem formulated as follows: for a given set of parameters, find u - ice velocity, h - ice thickness and x_g - the grounding line position, such that the momentum balance, the steady-state mass balance and all boundary conditions are simultaneously satisfied. I have confirmed that the obtained profiles were indeed steady-state solutions by running time-dependent simulations using these solutions as initial conditions. The difference between time-dependent solutions and steady-state solutions was on the order of 10^{-24} .

R1: *let's call them $h_{ss}(x)$ and $S_{ss}(x)$. $h_{ss}(x)$ is at the flotation thickness at the steady-state GL position x_{SS} . The model is then initialized at time $t = 0$ with a thickness ($h(x, t = 0)$) and surface profile ($S(x, t = 0)$) which result in an initial grounding-line position that's 1 km upstream of x_{SS} .*

The model is initialized at time $t = 0$ with an imposed grounding-line position that is 1 km upstream of x_{SS} ; no additional computations are made prior to start of time-dependent simulations.

R1: *I would like the author to provide the details of how she constructed the initial perturbed*

state $h(x, t = 0)$ from the steady-state profile $h_{SS}(x)$ – I presume that $h(x, t = 0) < h_{SS}(x)$ in order to move the grounding line upstream.

There was no additional construction of the initial perturbed state. At the first step of the time-dependent simulations with the prescribed grounding line position 1 km upstream of its steady-state location, the momentum balance and time-dependent mass balance (with $h_t \neq 0$) together with all boundary conditions, including the flotation condition ($h(x_g = x_{SS} - 1 \text{ km}, t = 0) = h_f$, where h_f is the flotation thickness) are solved simultaneously using a direct iterative solver. The initial guess for these iterations is the steady-state solutions h_{SS} and u_{SS} . Because the grounding line is upstream of its steady-state position and the bed is retrograde, $h(x_g = x_{SS} - 1 \text{ km}, t = 0) > h_{SS}(x_{SS})$. The converged solution is such that $h_t(x) \neq 0$, the configuration is out of equilibrium and the grounding line starts to move.

I hope this clarifies the initialization procedure for perturbation experiments. I will be glad to provide further details if needed.

2 Specific Points

R1: *A few minor suggestions:*

1. line 17: “its it interior” → “its interior”

Corrected.

R1: 2. line 31: “atmosphere” → “atmospheric”.

Corrected.

R1: 3. line 33: no comma after “Both”

Corrected.

R1: 4. line 41: “That previously” → “that the previously...”

Corrected.

R1: 5. line 82: “allows to construct...” → “allows construction of” or “allows us to construct”

Corrected.

R1: 6. line 104: “this grounding line position is stable” – Did you mean “unstable” (since it’s on a retrograde slope in Figure 1)?

No, I meant “stable”. Although the slope is retrograde, the ablation rate is large, and for the case

of the accumulation/ablation rate being only a function of x , this steady-state position is stable.

Reviewers' Comments:

Reviewer #1:

Remarks to the Author:

I'd like to thank the author for her reply, but I'm no closer to understanding how the perturbation experiments are set up than I was before (and since the article was unmodified for this, a reader would not be any more enlightened than I am).

Once again, given a steady state ice sheet configuration, I don't understand how things are set up so that "At $t=0$ the grounding line positions are set 1 km upstream of their steady-state positions"

For example, if I were doing this experiment, I would start with the steady-state thickness profile and then thin it in some way so that the resulting grounding line position is 1 km upstream of the steady-state position. Is that what you did?

I'm confused how one could be using the steady-state thickness profile, with the same bedrock elevation, *and* a GL position that's 1 km upstream of the steady-state position -- it's not possible to have two different GL positions with the same thickness and bedrock elevation fields. (are you simply just setting the basal friction to zero to mimic floating ice?) It's a minor point, but speaks to how one would be able to reproduce the experiments in this paper.

Reviewer 1

R1: *I'd like to thank the author for her reply, but I'm no closer to understanding how the perturbation experiments are set up than I was before (and since the article was unmodified for this, a reader would not be any more enlightened than I am).*

I am grateful to the Reviewer for their assessment of the manuscript and my earlier response.

R1: *Once again, given a steady state ice sheet configuration, I don't understand how things are set up so that "At $t=0$ the grounding line positions are set 1 km upstream of their steady-state positions" For example, if I were doing this experiment, I would start with the steady-state thickness profile and then thin it in some way so that the resulting grounding line position is 1 km upstream of the steady-state position. Is that what you did?*

I'm confused how one could be using the steady-state thickness profile, with the same bedrock elevation, and a GL position that's 1 km upstream of the steady-state position – it's not possible to have two different GL positions with the same thickness and bedrock elevation fields. (are you simply just setting the basal friction to zero to mimic floating ice?) It's a minor point, but speaks to how one would be able to reproduce the experiments in this paper.

To address the Reviewer's comment, I have added a description of the time-dependent simulations to the section 2 of the Supplementary Information (lines 46-60) and a new Supplementary Figure 4 that shows initial conditions for the time-dependent simulations. These conditions are obtained as a part of the time-dependent simulations at $t = 0$. The procedure is the following: The position of the grounding line is prescribed to be 1 km upstream of the steady-state position, *i.e.*, $x_g(t = 0) = x_{gss} - 1$ km, where x_{gss} is the position of the steady-state grounding line. On the domain $0 \leq x \leq x_g(t = 0)$ at $t = 0$ the time-dependent problem (15) (Supplementary Information, Section 2) is solved iteratively using the steady-state profiles of ice thickness $h(0 \leq x \leq x_g(t = 0))$ and velocity $u(0 \leq x \leq x_g(t = 0))$ as initial guesses for the iterative procedure. Iterations stop when the residual (the difference between the terms of the left-hand side and the right-hand side of the problem (15)) becomes smaller than the relative tolerance 10^{-6} . Figure 4 shows the obtained profiles of $h(t = 0)$ and $u(t = 0)$ on the domain with the displaced grounding line position ($0 \leq x \leq x_g(t = 0)$) in red and their steady-state profiles on the domain with the steady-state grounding line position ($0 \leq x \leq x_{gss}$) in blue.

With regards to reproducibility of the results, the manuscript complies with the Journal Data and Code availability policy and the results and the models will be publicly available at Zenodo. The Data and Code availability statements are lines 237-243 of the main text. Although Comsol™

is a commercial software, it has a free trial license and interested readers will be able to evaluate the model setup and run models themselves.

If these explanations are not sufficient I would be grateful to the Reviewer if they could identify what specific aspects are not clear, as I am at loss and prefer not to guess what they can be.

Reviewers' Comments:

Reviewer #1:

Remarks to the Author:

I'd like to thank the author for her additions to the supplementary material. I think I now understand how the perturbation experiment is set up. As before, I recommend that this article be published.

Reviewer 1

R1: *I'd like to thank the author for her additions to the supplementary material. I think I now understand how the perturbation experiment is set up. As before, I recommend that this article be published.*

I am grateful to the Reviewer for their assessment of the manuscript and their valuable suggestions that greatly improved its readability.